# An analytical process of spatial autocorrelation functions based on Moran's index

**Yanguang Chen** [ORCID] *

Department of Geography, College of Urban and Environmental Sciences, Peking University, Beijing, China

* chenyg@pku.edu.cn

## Abstract

A number of spatial statistic measurements such as Moran's $I$ and Geary's $C$ can be used for spatial autocorrelation analysis. Spatial autocorrelation modeling proceeded from the 1-dimension autocorrelation of time series analysis, with time lag replaced by spatial weights so that the autocorrelation functions degenerated to autocorrelation coefficients. This paper develops 2-dimensional spatial autocorrelation functions based on the Moran index using the relative staircase function as a weight function to yield a spatial weight matrix with a displacement parameter. The displacement bears analogy with the time lag in time series analysis. Based on the spatial displacement parameter, two types of spatial autocorrelation functions are constructed for 2-dimensional spatial analysis. Then the partial spatial autocorrelation functions are derived by using the Yule-Walker recursive equation. The spatial autocorrelation functions are generalized to the autocorrelation functions based on Geary's coefficient and Getis' index. As an example, the new analytical framework was applied to the spatial autocorrelation modeling of Chinese cities. A conclusion can be reached that it is an effective method to build an autocorrelation function based on the relative step function. The spatial autocorrelation functions can be employed to reveal deep geographical information and perform spatial dynamic analysis, and lay the foundation for the scaling analysis of spatial correlation.

## 1 Introduction

Measuring spatial autocorrelation is an important method for quantitative analyses in geography. This method can be treated as a cornerstone of spatial statistics. However, present spatial autocorrelation analysis has two significant shortcomings, which hinder its application scope and effect. First, in the theoretical aspect, the scaling property of geographical spatial autocorrelation has not been emphasized. Conventional mathematical modeling and quantitative analysis depend on characteristic scales. If and only if we find the valid characteristic scales such as determinate length, eigenvalue, and mean, will we be able to make effective mathematical models. If a geographical distribution is a scale-free distribution, no characteristic scale can be found, and the conventional mathematical methods will be ineffective. In this case, the

**Data Availability Statement:** All the data can be found in the Supporting Information files.

**Funding:** This research was sponsored by the National Natural Science Foundation of China (Grant No 41671167. See: http://isisn.nsfc.gov.cn/egrantweb/). The funders had no role in study

design, data collection and analysis, decision to publish, or preparation of the manuscript.

**Competing interests:** The authors have declared that no competing interests exist.

mathematical tools based on characteristic scales should be replaced by those based on scaling analysis [1]. Second, in the methodological aspect, the spatial displacement parameter, which is equivalent to the time lag parameter in time series analysis, has not been underlined in auto-correlation analyses. Thus, spatial autocorrelation function analysis in a strict sense has not been developed yet. The development path of spatial autocorrelation analysis in geography can be summarized as follows. First, generalizing Pearson's simple cross-correlation coefficient to time series analysis to yield a 1-dimensional temporal auto-correlation function (TACF) based on a time lag parameter [2–5]. Second, generalizing temporal auto-correlation function to ordered space series and substituting the time lag with spatial displacement to yield a 1-dimensional spatial auto-correlation function (SACF) [1]. Third, generalizing the 1-dimensional spatial auto-correlation function to a 2-dimensional spatial dataset and replacing the displacement parameters with the spatial weight matrix to yield a 2-dimensional spatial auto-correlation coefficient, which is termed Moran's index, or Moran's $I$ for short, in the literature [6–8]. In principle, a time lag parameter corresponds to a spatial displacement parameter, which in turn corresponds to the weight matrix. Where 1-dimensional autocorrelation analysis is concerned, a series of time lag parameters correspond to a series of spatial displacement parameters. However, only one spatial weight matrix can be taken into account in conventional autocorrelation modeling.

If the variable distance is adopted instead of the fixed distance to construct the spatial weight matrix, the spatial autocorrelation function analysis may be created. Many spatial statisticians have thought of this, and variable distance has been introduced into spatial autocorrelation analysis in many ways [8–13]. However, the introduction of variable distance is only one of the necessary conditions to advance the methods of spatial autocorrelation function analysis. To develop this methodological framework, a series of key problems needs to be solved. The problems include how to select the distance attenuation function, how to define the spatial weight matrix, and how to have the spatial autocorrelation function effectively correspond with the autocorrelation function in time series analysis. This paper develops 2-dimensional spatial autocorrelation functions based on Moran's index and the corresponding analytical process, laying the foundation for scaling analysis based on spatial autocorrelation. A set of ordered spatial weight matrixes are introduced into the spatial autocorrelation models to construct the 2-dimensional spatial autocorrelation function. Based on the 2-dimensional autocorrelation function, spatial scaling analysis may be made in addition to a spatial correlation analysis. The parts of the paper are organized as follows. In Section 2, two types of spatial autocorrelation functions based on Moran's index are established by using the relative staircase function as a weight function. Then the autocorrelation functions based on Moran's index are generalized to the autocorrelation functions based on Geary's coefficient and Getis's index. In Section 3, empirical analyses are made to show how to utilize the spatial autocorrelation functions to make empirical analyses of geographical phenomena. In Section 4, several questions related to this work are discussed. Finally, the discussion are concluded by outlining the main points of this study.

## 2 Theoretical results

### 2.1 Simplified expression of Moran's index

The first measurement of spatial autocorrelation was the well-known Moran's index, which is in fact a spatial autocorrelation coefficient (SACC). The formula of Moran's index bears a complicated form, but the expression can be simplified by means of a normalized matrix and a standardized vector. The formulae and expressions are not new in this subsection, but they are helpful for us to understand the new mathematical process shown in next subsection. Suppose

there are $n$ elements (e.g., cities) in a system (e.g., a network of cities) which can be measured by a variable (e.g., city size), $x$. In the literature, the global Moran's index can be expressed as

$$I = \frac{n \sum\limits_{i=1}^{n} \sum\limits_{j=1}^{n} v_{ij}(x_i - \mu)(x_j - \mu)}{\sum\limits_{i=1}^{n} \sum\limits_{j=1}^{n} v_{ij} \sum\limits_{i=1}^{n} (x_i - \mu)^2},$$ (1)

where $I$ denotes Moran's $I$, $x_i$ is a size measurement of the $i$th element in a geographical spatial system ($i = 1,2,...,n$), $\mu$ represents the mean of $x_i$, $v_{ij}$ refers to the elements in a spatial contiguity matrix (SCM), $V$. The symbols can be developed as follows

$$x = \begin{bmatrix} x_1 & x_2 & \cdots & x_n \end{bmatrix}^{\mathrm{T}},$$ (2)

$$\mu = \frac{1}{n} \sum_{i=1}^{n} x_i,$$ (3)

$$V = \begin{bmatrix} v_{11} & v_{12} & \cdots & v_{1n} \\ v_{21} & v_{22} & \cdots & v_{2n} \\ \vdots & \vdots & \ddots & \vdots \\ v_{n1} & v_{n2} & \cdots & v_{nn} \end{bmatrix} = [v_{ij}]_{n \times n}.$$ (4)

The formula of Moran's index can be simplified and re-express as a quadratic form [14]

$$I = z^{\mathrm{T}} W z,$$ (5)

in which $z$ denotes the standardized size vector based on population standard deviation, $W$ is the unitized spatial contiguity matrix (USCM), i.e., a spatial weight matrix (SWM), and superscript T indicates matrix or vector transposition. The standardized size vector is as follows

$$z = \frac{x - \mu}{\sigma},$$ (6)

where $\sigma$ refers to population standard deviation, which can be expressed as

$$\sigma = \left[\frac{1}{n} \sum_{i=1}^{n} (x_i - \mu)^2\right]^{1/2} = \left[\frac{1}{n} (x - \mu)^{\mathrm{T}} (x - \mu)\right]^{1/2}.$$ (7)

The spatial weight matrix, $W$, can be expressed as

$$W = \frac{V}{V_0} = \begin{bmatrix} w_{11} & w_{12} & \cdots & w_{1n} \\ w_{21} & w_{22} & \cdots & w_{2n} \\ \vdots & \vdots & \ddots & \vdots \\ w_{n1} & w_{n2} & \ddots & w_{nn} \end{bmatrix} = [w_{ij}]_{n \times n},$$ (8)

where

$$V_0 = \sum_{i=1}^{n} \sum_{j=1}^{n} v_{ij} \qquad (9)$$

represents the summation of the numeric value of matrix elements, and

$$w_{ij} = \frac{v_{ij}}{V_0} = \frac{v_{ij}}{\sum_{i=1}^{n} \sum_{j=1}^{n} v_{ij}}, \qquad (10)$$

denotes the unitized value of the $i$th row and the $j$th column in the weight matrix. Apparently, the matrix $W$ satisfies the following relation

$$\sum_{i=1}^{n} \sum_{j=1}^{n} w_{ij} = 1, \qquad (11)$$

which is termed the normalization condition and $W$ is termed the normalization matrix in the literature [14]. Besides the unitization indicated by Eq (11), the matrix has another two characteristics. One is symmetry, i.e., $w_{ij} = w_{ji}$; the other is zero diagonal elements, namely, $|w_{ii}| = 0$, which implies no self-correlation of an element with itself. The spatial contiguity matrix comes from the spatial distance matrix, which is a symmetric hollow matrix. The distance axiom determines the properties of spatial weight matrices [15].

Scientific description always relies heavily on a characteristic scale of a geographical system. A characteristic scale is a typical scale of a system which can be represented by a 1-dimensional measure. Thus, characteristic scales are usually termed *characteristic length* in the literature [16–20]. In mathematics, characteristic scales include determinate radius, side length, eigenvalue, average values, and standard deviation. An eigenvalue, if it does not depend on measurement scale, can be treated as a characteristic length. Therefore, eigenvalues and eigenvectors are important in spatial autocorrelation analyses [6,21–24]. Mathematical transformation can be employed to identify eigenvalues, and thus identify characteristic length of spatial autocorrelation. For a transformation **T**, a function $f(x)$ is an eigen function if it satisfies the following relation

$$\mathbf{T}(f(x)) = \lambda f(x), \qquad (12)$$

where $\lambda$ is the corresponding eigenvalue of the function. If **T** denotes a scaling transformation, the eigenvalue $\lambda$ will be associated with fractal dimension, including correlation dimension [16,18,19]. This relation can be generalized to matrix equations. It can be proved that Moran's index is the eigenvalue of generalized spatial correlation matrixes [14]. Based on the inner product of the standardized size vector, a Real Spatial Correlation Matrix (RSCM) can be defined as

$$M = nW = z^{\mathrm{T}}zW, \qquad (13)$$

where $n = z^{\mathrm{T}}z$ represents the *inner product* of $z$. Thus we have

$$Mz = nWz = z^{\mathrm{T}}zWz = Iz, \qquad (14)$$

which indicates that $I$ is the characteristic root of the polynomial equation proceeding from the determinant of the matrix $nW$, and $z$ is just the corresponding characteristic vector. Based

on the outer product of $z$, an Ideal Spatial Correlation Matrix (ISCM) can be defined as

$$M^* = zz^\mathrm{T} W, \tag{15}$$

where $zz^\mathrm{T}$ represents the *outer product* of the standardized size vector. Then we have

$$M^* z = zz^\mathrm{T} Wz = Iz, \tag{16}$$

which implies that $I$ is the largest eigenvalue of the generalized spatial correlation matrix $zz^\mathrm{T} W$, and $z$ is just the corresponding eigenvector of $zz^\mathrm{T} W$. This suggests that geographers have been taking advantage of the characteristic parameter for spatial analyses based on autocorrelation. Using Eqs (14) and (16), we can generate canonical Moran's scatterplots for local spatial analyses.

## 2.2 Standard spatial autocorrelation function based on Moran's *I*

Conventional mathematical modeling and quantitative analysis are based on characteristic scales. A mathematical model of a system is usually involved with three scales, and thus includes three levels of parameters. The first is the macro-scale parameter indicating environmental level, the second is the micro-scale parameter indicating the element level, and the third is the characteristic scale indicating the key level [25]. As indicated above, a characteristic scale is often called a characteristic length since it is always a 1-dimensional measure [17,19]. In geometry, a characteristic length may be the radius of a circle or the side length of a square; in algebra, a characteristic length may be the eigen values of a square matrix or characteristic roots of a polynomial; in probability theory and statistics, a characteristic length may be the mean value and standard deviation of a probability distribution. As demonstrated above, Moran's index is the eigenvalues of the generalized spatial correlation matrixes. Although the characteristic scales are expressed as radius, length of a side, eigenvalue, mean value, standard deviation, and so on, the reverse is not necessarily true. In other words, the radius, the side length, the eigenvalue, the mean value and the standard deviation do not necessarily represent a characteristic scale. If and only if a quantity can be objectively determined and its value does not depend on the scale of measurement, the quantity can be used to represent a characteristic length. Can Moran's index be evaluated uniquely and objectively under given spatio-temporal conditions? This is still a pending question in theoretical and quantitative geographies. To find the answer, we should calculate the Moran's index by means of different spatial scales.

Moran's index is a spatial autocorrelation coefficient, but it can be generalized to a spatial autocorrelation function (SACF). A spatial autocorrelation function is a set of a series of ordered autocorrelation coefficients. The spatial autocorrelation function can be derived from the proper spatial weight functions. Four types of spatial weight functions can be used to generate spatial contiguity matrixes: the inverse power function, negative exponential function, absolute staircase function, and relative staircase function [8,15,26–29]. Among these spatial weight functions, the relative staircase function is the most suitable one for constructing a spatial autocorrelation function. A relative staircase function can be expressed as

$$v_{ij}(r) = f(r) = \begin{cases} 1, & 0 < d_{ij} \leq r \\ 0, & d_{ij} > r, d_{ij} = 0 \end{cases}, \tag{17}$$

where $d_{ij}$ denotes the distance between locations $i$ and $j$, and $r$ represents the threshold value of spatial distance. In the literature, the threshold value $r$ is always represented by an average value, and is treated as a constant. However, a complex system often has no effective average value. In other words, complex systems are scale-free systems and have no characteristic scales.

In this case, the quantitative analysis based on characteristic scale should be replaced by scaling analysis. In fact, spatial statisticians and theoretical geographers have been aware of uncertainty of the threshold, $r$ [9–12]. Suppose that $r$ is a variable rather than a constant. The spatial contiguity matrix, Eq (4), should be rewritten as

$$V(r) = \begin{bmatrix} v_{11}(r) & v_{12}(r) & \cdots & v_{1n}(r) \\ v_{21}(r) & v_{22}(r) & \cdots & v_{2n}(r) \\ \vdots & \vdots & \ddots & \vdots \\ v_{n1}(r) & v_{n2}(r) & \cdots & v_{nn}(r) \end{bmatrix} = [v_{ij}(r)]_{n \times n}. \tag{18}$$

Accordingly, the spatial weight matrix, Eq (8), can be re-expressed as

$$W(r) = \frac{V(r)}{V_0(r)} = \begin{bmatrix} w_{11}(r) & w_{12}(r) & \cdots & w_{1n}(r) \\ w_{21}(r) & w_{22}(r) & \cdots & w_{2n}(r) \\ \vdots & \vdots & \ddots & \vdots \\ w_{n1}(r) & w_{n2}(r) & \ddots & w_{nn}(r) \end{bmatrix} = [w_{ij}(r)]_{n \times n}, \tag{19}$$

where

$$V_0(r) = \sum_{i=1}^{n} \sum_{j=1}^{n} v_{ij}(r) = e^{\mathrm{T}} V(r) e, \tag{20}$$

$$w_{ij}(r) = \frac{v_{ij}(r)}{\displaystyle\sum_{i=1}^{n} \sum_{j=1}^{n} v_{ij}(r)}. \tag{21}$$

In Eq (20), $e = [1\ 1\ \ldots\ 1]^{\mathrm{T}}$ refers to the "constant" vector with components $e_i = 1$ ($i = 1, \ldots, n$) [6], which is also termed the $n$-by-1 vector of ones [21]. The unitization property of spatial weight matrices remain unchanged, i.e.,

$$\sum_{i=1}^{n} \sum_{j=1}^{n} w_{ij}(r) = 1. \tag{22}$$

The global spatial autocorrelation function (SACF) based on the cumulative correlation can be defined as

$$I_c(r) = z^{\mathrm{T}} W(r) z, \tag{23}$$

in which $I_c(r)$ refers to cumulative ACF. Eq (23) comes from the global Moran's index and relative staircase function, Eqs (5) and (17).

The distance threshold is a type of displacement parameter, which correspond to the time lag parameter in the temporal autocorrelation models of time series analysis. In this framework, Moran's index is no longer a spatial autocorrelation coefficient. It becomes a function of spatial displacement $r$. By means of the spatial autocorrelation function, we can make quantitative analyses of geographical spatial dynamics. The distance threshold can be discretized as $r_k = r_0 + ks$, where $k = 1, 2, 3, \ldots, m$ represents natural numbers, $s$ refers to step length, and $r_0$ is a constant. Empirically, the distance threshold comes between the minimum distance and the maximum distance, namely, $\min(d_{ij}) \leq r \leq \max(d_{ij})$. The global spatial autocorrelation function

(SACF) based on density correlation can be computed by

$$I_d(r) = \begin{cases} I(r_k) = z^{\mathrm{T}} W(r_1) z, & k = 1 \\ \Delta I(r_k) = z^{\mathrm{T}} W(r_k) z - z^{\mathrm{T}} W(r_{k-1}) z, & k > 1 \end{cases}, \tag{24}$$

which indicates that the density correlation function is the differences of cumulative correlation function.

## 2.3 Generalized spatial autocorrelation function based on Moran's *I*

In the above defined spatial autocorrelation function, each value represents an autocorrelation coefficient. In other words, if a distance threshold value *r* is given, then we have a standard Moran's index. Spatial autocorrelation analysis originated from time series analysis. However, this kind of autocorrelation function does not bear the same structure with the temporal autocorrelation function in time series analysis. If we construct a "weight matrix" to compute the autocorrelation function of a time series, the "weight matrix" is a quasi-unitized matrix instead of a strict unitized matrix. Actually, by analogy with the temporal autocorrelation function, we can improve the spatial autocorrelation function by revising the spatial weight matrix. The key lies in Eq (20). According to the property of the spatial contiguity matrix based on the relative staircase function, the maximum of $V_0(r)$ is

$$\max(V_0(r)) = \lim_{r \to \max(d_{ij})} \sum_{i=1}^{n} \sum_{j=1}^{n} v_{ij}(r) = n(n-1). \tag{25}$$

Thus the spatial weight matrix, Eq (19), can be revised as

$$W^*(r) = \frac{1}{n(n-1)} V(r) = \begin{bmatrix} w_{11}^*(r) & w_{12}^*(r) & \cdots & w_{1n}^*(r) \\ w_{21}^*(r) & w_{22}^*(r) & \cdots & w_{2n}^*(r) \\ \vdots & \vdots & \ddots & \vdots \\ w_{n1}^*(r) & w_{n2}^*(r) & \ddots & w_{nn}^*(r) \end{bmatrix} = [w_{ij}^*(r)]_{n \times n}. \tag{26}$$

Based on Eq (26), the spatial autocorrelation function can be re-defined as

$$I^*(r) = z^{\mathrm{T}} W^*(r) z = \frac{1}{n(n-1)} z^{\mathrm{T}} V(r) z, \tag{27}$$

which bears a strict analogy with the temporal autocorrelation function of time series analysis. The difference between Eqs (23) and (27) is as follows, for $I(r)$, $V_0(r)$ is a variable which depends on the distance threshold *r*, while for $I^*(r)$, $V_0(r)$ is a constant which is independent of *r*. In this case, the spatial weight matrix does not always satisfy the unitization condition, and we have an inequality as below

$$\sum_{i=1}^{n} \sum_{j=1}^{n} w_{ij}^*(r) \le 1. \tag{28}$$

This implies that the summation of the elements in $W^*(r)$ is equal to or less than 1.

## 2.4 Partial spatial autocorrelation function based on Moran's *I*

Autocorrelation coefficients reflect both direct correlation and indirect correlation between the elements in a sample. If we want to measure the pure direct autocorrelation and neglect

the indirect autocorrelation, we should compute the partial autocorrelation coefficients. A set of ordered partial autocorrelation coefficients compose an autocorrelation function. Generally speaking, the partial spatial autocorrelation function (PSACF) should be calculated by the SACF based on density correlation function. In fact, we transform the spatial autocorrelation functions into the partial autocorrelation functions by means of the Yule-Walker recursive equation [30,31]

$$
\begin{bmatrix} I_1 \\ I_2 \\ I_3 \\ \vdots \\ I_m \end{bmatrix} = \begin{bmatrix} 1 & I_1 & I_2 & \cdots & I_{m-1} \\ I_1 & 1 & I_1 & \cdots & I_{m-2} \\ I_2 & I_1 & 1 & \cdots & I_{m-3} \\ \vdots & \vdots & \vdots & \ddots & \vdots \\ I_{m-1} & I_{m-2} & I_{m-3} & \cdots & 1 \end{bmatrix} \cdot \begin{bmatrix} H_1 \\ H_2 \\ H_3 \\ \vdots \\ H_m \end{bmatrix}. \tag{29}
$$

where $I_k$ denotes the $k$th order autocorrelation coefficient, and the parameter $H_k$ is the corresponding auto-regression coefficients. The Yule-Walker equation associates the autocorrelation with the auto-regression equations. The last auto-regression coefficient, $H_m$, is equal to the $m$th order partial autocorrelation coefficient ($k$ = 1, 2, 3,..., $m$). If $m$ = 1, we have the first-order partial autocorrelation coefficient, which can be given by

$$
[J_1] = [1] \cdot [H_1] = [H_1], \tag{30}
$$

in which $J_1 = H_1$ is the first-order partial autocorrelation coefficient. If $m$ = 2, we have the second-order partial autocorrelation coefficient, which can be given by the following matrix equation

$$
\begin{bmatrix} I_1 \\ I_2 \end{bmatrix} = \begin{bmatrix} 1 & I_1 \\ I_1 & 1 \end{bmatrix} \cdot \begin{bmatrix} H_1 \\ H_2 \end{bmatrix}, \tag{31}
$$

where $J_2 = H_2$ is the second-order partial autocorrelation coefficient. If $m$ = 3, we have the third-order partial autocorrelation coefficient, which can be given by

$$
\begin{bmatrix} I_1 \\ I_2 \\ I_3 \end{bmatrix} = \begin{bmatrix} 1 & I_1 & I_2 \\ I_1 & 1 & I_1 \\ I_2 & I_1 & 1 \end{bmatrix} \cdot \begin{bmatrix} H_1 \\ H_2 \\ H_3 \end{bmatrix}, \tag{32}
$$

in which $J_3 = H_3$ is the third-order partial autocorrelation coefficient. Among these matrix equations, Eq (30) is a special case. It suggests that the first-order autocorrelation coefficient equals the first-order partial autocorrelation coefficient, which in turn equals the first-order auto-regression coefficient. For Eqs (31) and (32), we can calculate the autoregressive coefficient by means of finding the inverse matrix of the autocorrelation coefficient matrix. The last autoregressive coefficient gives the partial autocorrelation coefficient value. The others can be obtained by analogy. Applying Eqs (23) to (29) yields the partial spatial autocorrelation function based on cumulative correlation, and applying Eqs (24) to (29) yields the partial spatial autocorrelation function based on density correlation. For $n$ spatial elements, the correlation number is $n^*n$. Thus, based on significance level of 0.05, the standard deviation of the spatial SACF and PSACF can be estimated by the formula, $1/n$.

## 2.5 Spatial autocorrelation functions based on Geary's *C* and Getis' *G*

In order to carry out more comprehensive spatial autocorrelation analysis, the spatial autocorrelation functions can be extended to more spatial statistical measurements. Besides Moran's index, the common spatial autocorrelation measurements include Geary's coefficient and Getis-Ord's index [11,32]. The former is often termed Geary's *C*, and the latter is also termed Getis's *G* for short in the literature [6,8,33]. It is easy to generalize the 2-dimensional spatial autocorrelation functions to Geary's coefficient according to the association of Moran's index with Geary's coefficient. In theory, Geary's coefficient is equivalent to Moran's index, but in practice, the former is based on a sample, while the latter is based on the population [14,15]. Geary's coefficient can be expressed in the following form [32]:

$$
C = \frac{(n-1)\sum_{i=1}^{n}\sum_{j=1}^{n}v_{ij}(x_i-x_j)^2}{2\sum_{i=1}^{n}\sum_{j=1}^{n}v_{ij}\sum_{i=1}^{n}(x_i-\bar{x})^2} = \frac{(n-1)\sum_{i=1}^{n}\sum_{j=1}^{n}w_{ij}(x_i-x_j)^2}{2\sum_{i=1}^{n}(x_i-\bar{x})^2}. \tag{33}
$$

Based on matrix and vector, Eq (33) can be simplified to the following form

$$
C = \frac{n-1}{n}(e^{\mathrm{T}}Wz^2 - z^{\mathrm{T}}Wz) = \frac{n-1}{n}(e^{\mathrm{T}}Wz^2 - I), \tag{34}
$$

where $e = [1\ 1\ \ldots\ 1]^{\mathrm{T}}$, and $z^2 = D_{(z)}z = [z_1^2\ z_2^2\ \ldots\ z_n^2]^{\mathrm{T}}$. Here $D_{(z)}$ is the diagonal matrix consisting of the elements of $z$. Eq (34) gives the exact relation between Moran's index $I$ and Geary's coefficient $C$. Introducing the spatial displacement parameter into Eq (34) yields two autocorrelation functions as follows

$$
C(r) = \frac{n-1}{n}[e^{\mathrm{T}}W(r)z^2 - I(r)] = \frac{n-1}{n}[e^{\mathrm{T}}W(r)z^2 - z^{\mathrm{T}}W(r)z], \tag{35}
$$

$$
C^*(r) = \frac{n-1}{n}[e^{\mathrm{T}}W^*(r)z^2 - I^*(r)] = \frac{n-1}{n}[e^{\mathrm{T}}W^*(r)z^2 - z^{\mathrm{T}}W^*(r)z]. \tag{36}
$$

Clearly, Eq (35) is based on the standard unitized spatial weight matrix, corresponding to Eq (23), while Eq (36) is based on the quasi-unitized spatial weight matrix, corresponding to Eq (27).

Further, the analytical process of spatial autocorrelation functions can be generalized to Getis' index. Based on the unitized size vector, the formula of Getis' index can be simplified to the form similar to the new expression of Moran's index, Eq (5). Generally speaking, Getis's index, *G*, is expressed as below [11]:

$$
G = \frac{\sum_{i=1}^{n}\sum_{j=1}^{n}w_{ij}x_ix_j}{\sum_{i=1}^{n}\sum_{j=1}^{n}x_ix_j}. \tag{37}
$$

The notation is the same as those in Eq (1). Suppose that for the outer product $x_ix_j$, $i = j$ can be kept, but for weight, $w_{ij}$, $i = j$ is rejected. Using unitized matrix and unitized vector, we can rewrite Eq (37) in the following simple form [34]

$$
G = y^{\mathrm{T}}Wy, \tag{38}
$$

where $y = x/S = [y_1, y_2, \ldots, y_n]^T$ represents the unitized vector of $x$. The elements of $y$ is defined as below:

$$y_i = \frac{x_i}{S} = x_i / \sum_{i=1}^{n} x_i = \frac{x_i}{n\bar{x}}, \tag{39}$$

where the sum of $x$ is

$$S = \sum_{i=1}^{n} x_i. \tag{40}$$

Now, introducing the spatial displacement parameter $r$ into Eq (38) produces two autocorrelation functions as below

$$G(r) = y^T W(r) y, \tag{41}$$

$$G^*(r) = y^T W^*(r) y. \tag{42}$$

Similar to Eqs (35), (36) and (41) is based on the standard unitized spatial weight matrix, corresponding to Eq (23), while Eq (42) is based on the quasi-unitized spatial weight matrix, corresponding to Eq (27).

## 3 Materials and methods

### 3.1 Data and analytical approach

The analytical process of spatial autocorrelation functions can be used to research the dynamic spatial structure of China's system of cities. As a case demonstration of our methodology, only the capital cities of the 31 provinces, autonomous regions, and municipalities directly under the Central Government of China are taken into consideration for simplicity. The urban population is employed as a size measurement, while the distances by train between any two cities act as a spatial contiguity measurement. The census data of the urban population in 2000 and 2010 are available from the Chinese website, and the railroad distance matrix can be found in many Chinese traffic atlases [14,15,34]. The cities of Haikou and Lhasa are not taken into account in this study. Located in Hainan Island, Haikou is the capital of Hainan Province and do not be linked to other cities by railway (lack of traffic mileage data of Haikou to other cities in the atlas). Lhasa is the capital of the Tibetan Autonomous Region, located on the Qinghai Tibet Plateau. Because of the plateau climate, Lhasa is loosely connected with other mainland cities of China. In fact, the Qinghai Tibet railway is still under construction. Therefore, 29 Chinese capital cities are actually included in the datasets. In this case, the spatial sample size of the urban population is $n = 29$ (See S1 and S2 Files). First of all, the staircase function was used to determine a spatial contiguity matrix based on a threshold distance $r$. Then, the three-step method was employed to calculate Moran's index and Getis-Ord's index [14,34]. The process is as follows. Step 1: standardizing the size vector $x$ yields standardized size vector $z$. Step 2: unitizing the spatial contiguity matrix based on a threshold distance $r$ yields a spatial weight matrix $W(r)$. Step 3: computing Moran's index using Eq (23). The values of Moran's index can be converted into the corresponding values of Geary's coefficient with Eq (34). As for Getis's index, the standardized size variable $z$ should be replaced by the utilized size variable $y$, and the formula is Eq (38). Changing the distance threshold $r$ value yields different values of Moran's index, Geary's coefficient, and Getis's index. Thus we have spatial autocorrelation functions based on cumulative distributions (correlation cumulation). The differences of cumulative distributions give the spatial autocorrelation functions base on density distribution (correlation

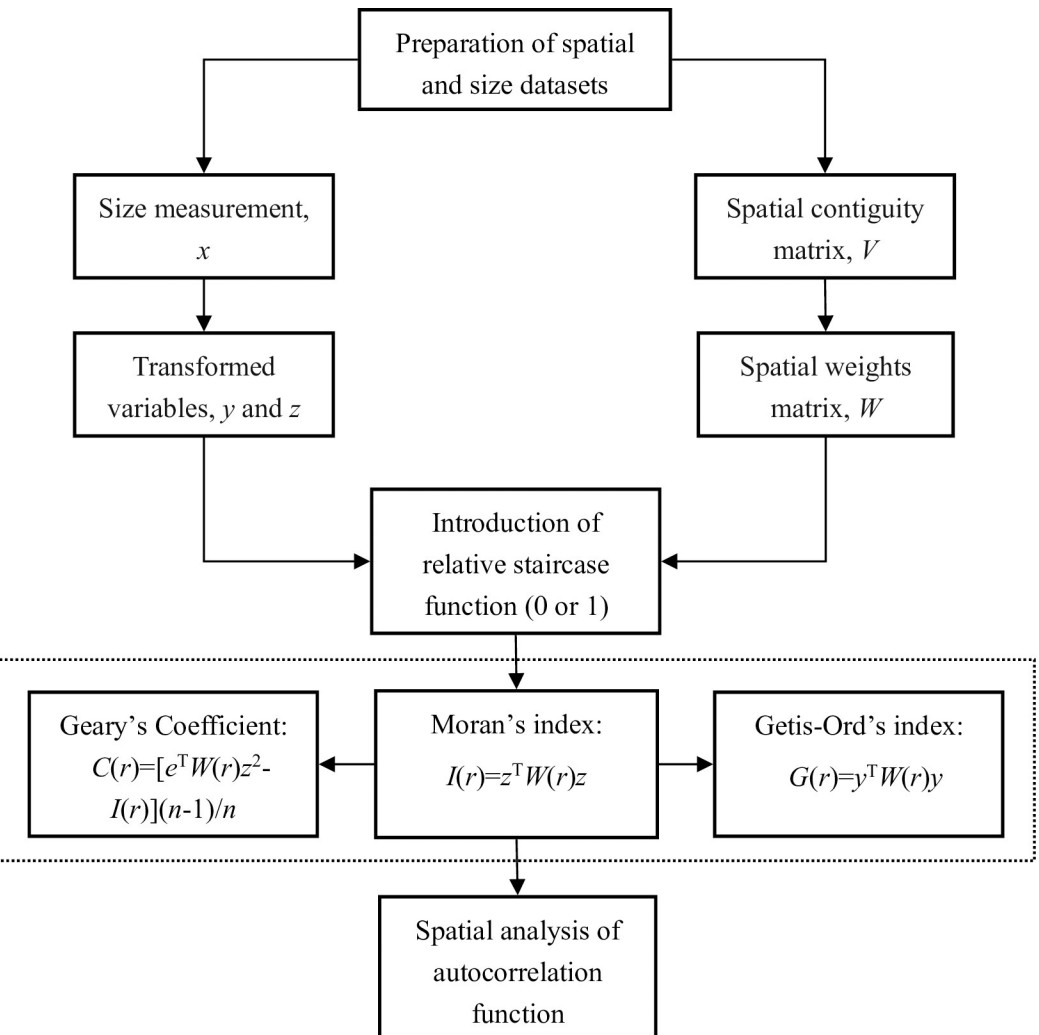

**Fig 1. A flow chart of data processing, parameter estimation, and spatial autocorrelation function analyses.** (**Note**: The analytical process is based on the improved mathematical expressions of Moran's *I*, Geary's *C* and Getis-Ord's *G*. Compared with *x*, *y* represents the unitized size variable, and *z*, the standardized variable).

density). The main calculation process of spatial autocorrelation functions can be illustrated as follows (Fig 1). It is easy to realize the whole calculation process by computer programming (See S3 File).

The spatial analytical process and results rely heavily on the definition and structure of the spatial contiguity matrix. Two aspects of factors in its structure significantly impact the analytical process. One is the diagonal elements, and the other is the sum of the spatial contiguity values. (1) *Diagonal elements of spatial contiguity matrix*. For conventional spatial autocorrelation analysis, the diagonal elements should be removed; while for spatial correlation dimension analysis, the diagonal elements must be taken into account. As a matter of fact, theoretical geographers and spatial statisticians have taken into account the diagonal elements for the spatial weight matrix [11,13]. Where generalized spatial autocorrelation functions are concerned, the diagonal elements of the spatial contiguity matrix should not be zero. As for special fractal analysis, the diagonal element can be overlooked. (2) *The sum of spatial contiguity matrix*. For the theoretical spatial autocorrelation function, the sum varies with the yardstick length.

**Table 1. Four possible types of calculation approaches to spatial autocorrelation functions based on different diagonal elements and means of spatial contiguity matrix.**

| | Variable sum of distance matrix [V] | Fixed sum of distance matrix [F] |
|---|---|---|
| **All elements (including diagonal elements) [D]** | [D+V] Generalized Moran's function, $I^*(r)$; the sum of spatial contiguity matrix elements is $N(r)$ | [D+F] Generalized Moran's function, $I_f^*(r)$, the sum of spatial contiguity matrix elements is $N^2$ |
| **Partial elements (excluding diagonal elements) [N]** | [N+V] Conventional Moran's function, $I(r)$; the sum of spatial contiguity matrix elements is $N(r)$-$N$ | [N+F] Conventional Moran's function, $I_f(r)$; the sum of spatial contiguity matrix elements is $N(N$-$1)$ |
| **Application direction** | Theoretical study and fractal analysis | Practical study and spatial autocorrelation analysis |

However, for a practical spatial autocorrelation function, the sum of spatial contiguity matrix should be fixed to the original sum value. Different diagonal elements plus different definitions of the sum of the spatial contiguity matrix lead to four approaches to autocorrelation function analyses (Table 1).

## 3.2 Empirical analysis of spatial autocorrelation

The spatial contiguity matrix in spatial autocorrelation analysis bears analogy with the time lag parameter in time series analysis. Normalizing a spatial contiguity matrix yields a spatial weight matrix, and the former is equivalent to the latter. Therefore, the spatial weight matrix is always confused with the spatial contiguity matrix in the literature. A spatial contiguity matrix is based on the generalized distance matrix and must satisfy the axiom of distance. This suggests that a spatial weight matrix must be a nonnegative definite symmetric matrix. It is easy to generate a spatial contiguity matrix by using a weight function [26,29]. For $n$ elements in a geographical system, a spatial contiguity matrix, $V(r)$, can be produced by means of Eq (17). Normalizing the matrix $V(r)$ yields the spatial weight matrix $W(r)$. Changing the distance threshold, i.e., the yardstick length $r$, results in a different weight matrix $W(r)$, and thus results in a different Moran's index $I(r)$. A set of Moran's index values compose Moran's function. The spatial autocorrelation functions based on cumulative correlation can be converted into those based on density correlation by using difference method. Moran's autocorrelation function can be turned into Moran's partial autocorrelation function through the Yule-Walker recursive equation. The results are tabulated as below (Table 2). The significance of the autocorrelation function can be judged by the double values of positive and negative standard errors. The standard error value can be estimated with the reciprocal of the square root of the number of sampling points [35]. For spatial autocorrelation, if $n$ geographical elements are taken into account, the maximum number of sampling points can be treated as $n^2$, and thus, based on the significance level $\alpha = 0.05$, the standard errors can is about $1/n$. Adding the positive and negative double standard error lines to correlograms yields what is called "two-standard-error bands" [35] (Figs 2–5).

First of all, let us investigate the generalized spatial autocorrelation function based on cumulative correlation and the corresponding partial spatial autocorrelation function. These functions reflect the distance decay effect. The generalized autocorrelation function is based on the spatial contiguity matrix with nonzero diagonal elements, and the sum of the matrix elements is fixed to a constant $n^2 = 29^*29 = 841$. That is to say, for every yardstick $r$, the number 841 is employed to divide the sum of the spatial contiguity matrix elements, and the normalized results represent the spatial weight matrix. This autocorrelation coefficient includes two parts: one is $i$ correlates $i$ and $j$ correlates $j$ (based on diagonal elements), and the other, $i$ correlates $j$ and $j$ correlates $i$ (based on the elements outside the diagonal of the matrix). The dynamic properties of the generalized spatial autocorrelation are as below. (1) With the increase of threshold distance, both the autocorrelation function and partial autocorrelation function show wave attenuation.

**Table 2. Datasets for spatial autocorrelation function (ACF) and partial spatial autocorrelation function (PACF) based on Moran's index (partial results).**

| Scale | 2000 (Fifth census data) | | | | 2010 (Sixth census data) | | | |
|---|---|---|---|---|---|---|---|---|
| r (km) | D+F | | N+F | | D+F | | N+F | |
| | ACF $\overset{*}{I}(r)$ | PACF $\overset{*}{J}(r)$ | ACF $\Delta I(r)$ | PACF $\Delta J(r)$ | ACF $\overset{*}{I}(r)$ | PACF $\overset{*}{J}(r)$ | ACF $\Delta I(r)$ | PACF $\Delta J(r)$ |
| 150 | 0.0384 | 0.0384 | 0.0040 | 0.0040 | 0.0412 | 0.0412 | 0.0069 | 0.0069 |
| 250 | 0.0372 | 0.0357 | -0.0013 | -0.0013 | 0.0424 | 0.0408 | 0.0012 | 0.0012 |
| 350 | 0.0344 | 0.0318 | -0.0028 | -0.0028 | 0.0404 | 0.0372 | -0.0021 | -0.0021 |
| 450 | 0.0309 | 0.0273 | -0.0036 | -0.0036 | 0.0375 | 0.0329 | -0.0030 | -0.0029 |
| 550 | 0.0291 | 0.0248 | -0.0019 | -0.0019 | 0.0334 | 0.0278 | -0.0043 | -0.0042 |
| 650 | 0.0264 | 0.0216 | -0.0027 | -0.0027 | 0.0327 | 0.0264 | -0.0007 | -0.0007 |
| 750 | 0.0254 | 0.0202 | -0.0011 | -0.0011 | 0.0294 | 0.0225 | -0.0034 | -0.0034 |
| 850 | 0.0176 | 0.0120 | -0.0081 | -0.0081 | 0.0201 | 0.0127 | -0.0097 | -0.0096 |
| 950 | 0.0199 | 0.0145 | 0.0024 | 0.0024 | 0.0230 | 0.0159 | 0.0031 | 0.0032 |
| 1050 | 0.0109 | 0.0054 | -0.0094 | -0.0094 | 0.0121 | 0.0049 | -0.0114 | -0.0114 |
| 1150 | 0.0119 | 0.0069 | 0.0011 | 0.0011 | 0.0117 | 0.0052 | -0.0004 | -0.0003 |
| 1250 | 0.0203 | 0.0157 | 0.0087 | 0.0086 | 0.0143 | 0.0085 | 0.0027 | 0.0026 |
| 1350 | 0.0125 | 0.0076 | -0.0080 | -0.0082 | 0.0110 | 0.0055 | -0.0034 | -0.0035 |
| 1450 | 0.0122 | 0.0075 | -0.0003 | -0.0003 | 0.0078 | 0.0027 | -0.0033 | -0.0034 |
| 1550 | 0.0301 | 0.0257 | 0.0185 | 0.0185 | 0.0270 | 0.0227 | 0.0199 | 0.0198 |
| 1650 | 0.0222 | 0.0167 | -0.0082 | -0.0084 | 0.0214 | 0.0161 | -0.0058 | -0.0062 |
| 1750 | 0.0176 | 0.0115 | -0.0048 | -0.0047 | 0.0170 | 0.0112 | -0.0045 | -0.0045 |
| 1850 | 0.0255 | 0.0193 | 0.0082 | 0.0082 | 0.0224 | 0.0163 | 0.0056 | 0.0055 |
| 1950 | 0.0195 | 0.0126 | -0.0062 | -0.0062 | 0.0185 | 0.0119 | -0.0040 | -0.0039 |
| 2050 | 0.0243 | 0.0173 | 0.0050 | 0.0050 | 0.0224 | 0.0155 | 0.0040 | 0.0041 |
| 2150 | 0.0116 | 0.0040 | -0.0131 | -0.0132 | 0.0098 | 0.0022 | -0.0131 | -0.0132 |
| 2250 | 0.0029 | -0.0043 | -0.0090 | -0.0087 | 0.0003 | -0.0071 | -0.0098 | -0.0095 |
| 2350 | 0.0157 | 0.0097 | 0.0133 | 0.0134 | 0.0128 | 0.0068 | 0.0129 | 0.0134 |
| 2450 | 0.0034 | -0.0030 | -0.0128 | -0.0133 | 0.0032 | -0.0028 | -0.0100 | -0.0106 |
| 2550 | 0.0139 | 0.0087 | 0.0109 | 0.0114 | 0.0135 | 0.0086 | 0.0107 | 0.0112 |
| 2650 | 0.0078 | 0.0026 | -0.0064 | -0.0066 | 0.0076 | 0.0028 | -0.0061 | -0.0064 |
| 2750 | 0.0039 | -0.0013 | -0.0040 | -0.0046 | 0.0025 | -0.0021 | -0.0053 | -0.0056 |
| 2850 | 0.0019 | -0.0025 | -0.0021 | -0.0014 | 0.0006 | -0.0032 | -0.0020 | -0.0015 |
| 2950 | 0.0022 | -0.0016 | 0.0004 | 0.0000 | 0.0016 | -0.0015 | 0.0011 | 0.0008 |
| 3050 | -0.0046 | -0.0085 | -0.0071 | -0.0077 | -0.0046 | -0.0076 | -0.0064 | -0.0069 |

**Note:** (1) Only partial results are tabulated. See the Supporting Information files for more results. (2) **D** implies that diagonal elements are taken into account, **N** denotes that diagonal elements are deleted, and **F** means fixed mean values of spatial contiguity matrix elements. (3) ACF represents spatial autocorrelation function, and PACF refers to partial spatial autocorrelation function. (4) For [D+F] type, ACF and PACF are based on cumulative correlation, while for [N+F] type, ACF and PACF are based on density correlation. (5) The unit of distance is kilometer (km).

(2) The shape of the autocorrelation function curve is similar to that of partial autocorrelation function curve. (3) From 2000 to 2010, the shape of the autocorrelation function curve showed no significant change (Figs 2 and 3). Therefore, it can be concluded that the spatial relationship between Chinese cities is relatively stable, and the direct relationship between different cities is relatively weak. From about 2750 km to 2950 km, the positive correlation becomes weak and even turns to negative correlation. This indicates that the distance 150 to 2950 km is a significant correlation range in the spatial distribution of Chinese cities.

Secondly, let us examine the standard spatial autocorrelation function based on density correlation and partial autocorrelation function. These functions reflect the spatial transition and

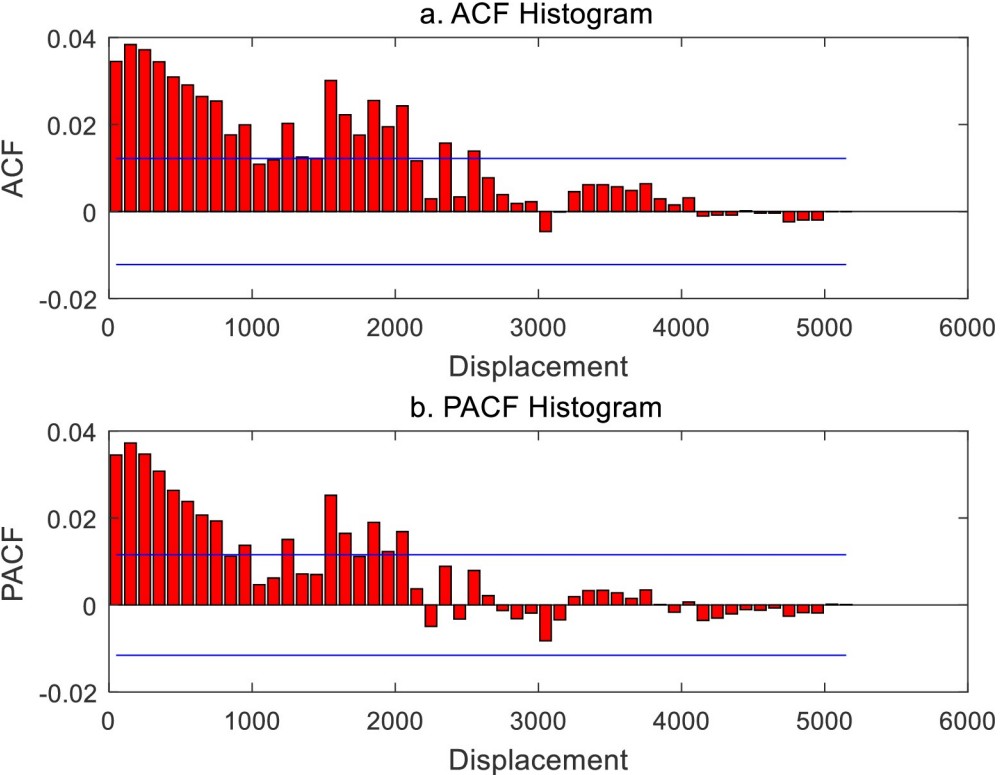

**Fig 2. Spatial autocorrelation function and partial autocorrelation function of Chinese cities based on generalized Moran's index and correlation cumulation (2000).** (**Note**: The blue lines in the histograms are termed "two-standard-error bands", according to which we can know whether or not there is significant difference between ACF or PACF values and zero. The same below.).

oscillation between positive autocorrelation and negative autocorrelation. The standard auto-correlation function is based on the spatial contiguity matrix with zero diagonal elements, and the sum of the matrix elements is fixed to a constant $(n\text{-}1)n = 28*29 = 812$ for different yard-stick $r$. That is, the sum of spatial contiguity matrix elements is divided by the number 812, and the normalized results serve as the spatial weight matrix. This autocorrelation coefficient includes only one part, namely, $i$ correlates $j$ and $j$ correlates $i$ (based on the elements outside the diagonal of the matrix). Another part, i.e., $i$ correlates $i$ and $j$ correlates $j$ (based on diagonal elements), is ignored. To reflect the sensitivity of spatial correlation, the cumulative autocorre-lation functions are transformed into density autocorrelation functions. The dynamic proper-ties of the standard spatial autocorrelation are as follows. (1) If the distance is too short and too remote, the autocorrelation is very weak. Only when the distance is proper is the autocorrela-tion significant. (2) The pattern of indirect correlation reflected by the autocorrelation func-tion looks very like the pattern of direct correlation reflected by the partial autocorrelation function. (3) From 2000 to 2010, the autocorrelation and partial autocorrelation patterns have no significant change (Figs 4 and 5).

However, if we calculate the ratio of the spatial autocorrelation function and the partial spa-tial autocorrelation function, we will find the inherent regularities. When the curves of auto-correlation function and partial autocorrelation function fluctuate sharply, their ratio is very stable. On the contrary, when the autocorrelation function and partial autocorrelation func-tion seem to be stable, their ratio changes sharply (Fig 6). Combining the results of two auto-correlation functions, the characteristic correlation ranges can be obtained. This can be treated

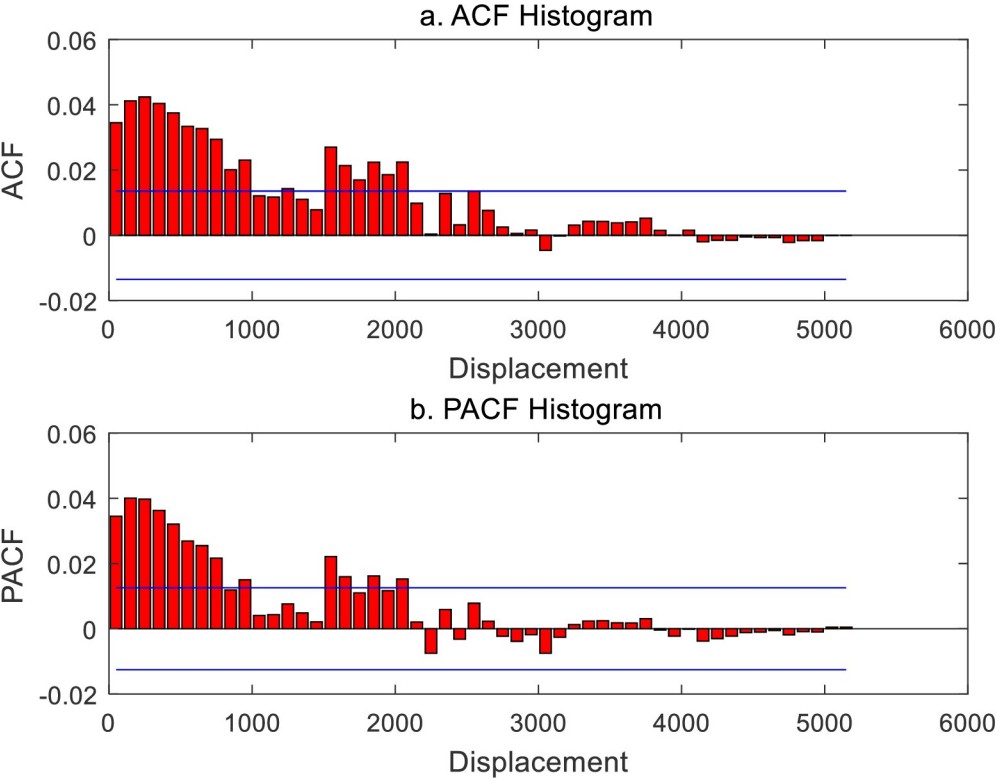

**Fig 3. Spatial autocorrelation function and partial autocorrelation function of Chinese cities based on generalized Moran's index and correlation cumulation (2010).**

as the scaling ranges of spatial correlation of Chinese cities. For 2000, the scaling range comes between about 250 km and 2850 km. For 2010, the scaling range comes between about 250 km and 3350 km. In fact, based on the scales ranging 250 to 2750, a scaling exponent, spatial correlation dimension, can be revealed from the relationships between yardstick lengths and corresponding correlation numbers of cities, and the result is about $D = 1.7$.

If a geographical system has a typical scale, we can utilize the parameter indicating characteristic length to perform spatial analysis. In this case, we can find the characteristic scale of spatial autocorrelation [8]. In contrast, if the autocorrelation coefficient values depend on measurement scale and no determinate typical value of Moran's index can be found, we meet a scale-free system, and the characteristic length should be replaced by a scaling process. Scaling range is important for geographical spatial analysis from the perspective of spatial complexity. An interesting finding in this work is that, within the scaling range, all the autocorrelation measurements based on density correlation change sharply over distance, but the ratio of the autocorrelation function to the corresponding partial autocorrelation function is very stable. Besides Moran's index, the changing feature of spatial autocorrelation can be reflected by Moran's scatterplots. Based on spatial autocorrelation functions, a series of canonical Moran's scatterplots can be drawn by means of Eqs (14) and (16). These graphs can reflect the positive and negative alternation process and local characteristics of spatial autocorrelation (Fig 7).

By using spatial ACF and PACF, we can show that the spatial autocorrelation characteristics of Chinese cities are as follows. First, the spatial autocorrelation of population among the principal cities in China is weak. Most absolute values of autocorrelation coefficients are less than the double value of the standard error, 1.96/29, i.e., 0.0676. The reason lies in two aspects: one

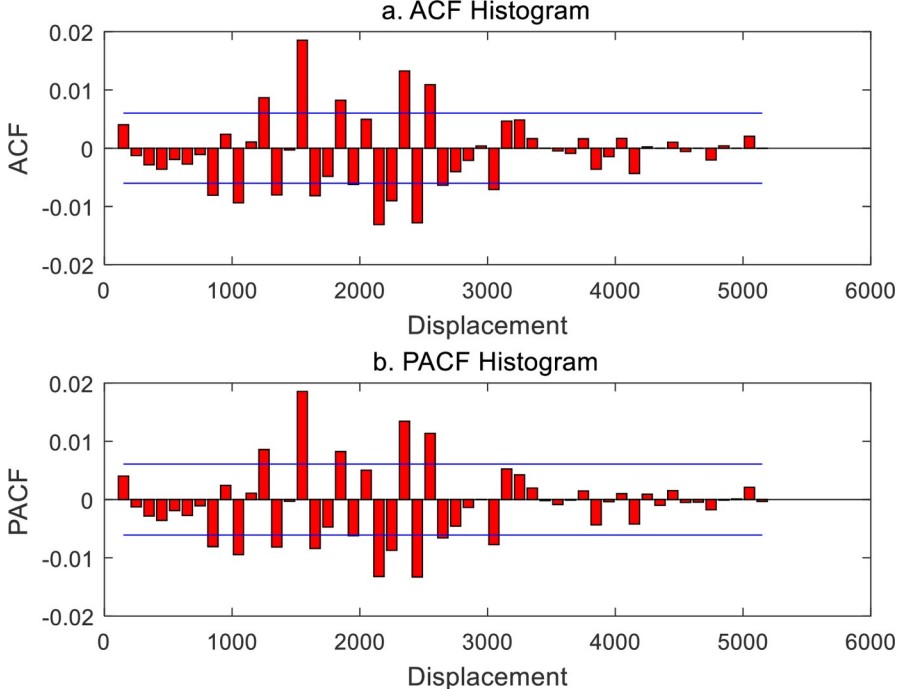

**Fig 4. Spatial autocorrelation function and partial autocorrelation function of Chinese cities based on conventional Moran's index and correlation density (2000).**

is the large territory of China, and the other is the strict registered residence management system. Therefore, on the national dimension, population migration between large cities is not free. Second, population flow among Chinese cities takes on self-correlation, namely, a city influences itself. As indicated above, the self-correlation is reflected by the diagonal elements. If the diagonal elements are taken into account, there is no significant difference between the spatial ACF and to the spatial PACF. On the other, the correlogram based on zero diagonal weight matrix differ significantly from that based on nonzero diagonal weight matrix. This suggests that the diagonal elements indicative of self-correlation play a significant part in the calculation of spatial ACF and PACF. Third, the spatial autocorrelation fluctuates sharply within a certain scale range. This can be seen by the standard spatial ACF and PACF. When the distance is less than 2750 km, the spatial autocorrelation changes significantly with distance, but when the distance is more than 3350 km, the spatial autocorrelation does not change significantly with distance. The maximum effective distance of urban spatial correlation in China seems to be about 3000 km (2750–3350 km). This can be treated as a characteristic length of spatial autocorrelation of Chinese cities.

### 3.3 Scaling analysis of spatial autocorrelation

The autocorrelation functions based on Moran's index involves negative values, and cannot be directly associated with scaling relation. The solution to a scaling equation is always a power law. So the power law is the basic mark of scaling in empirical studies. In Eq (12), if **T** represents a contraction-dilation transformation, and a function satisfies Eq (12), we will say it follows the scaling law. The values of Geary's coefficient and Getis-Ord's index are greater than 0 in empirical studies and may follow a power law. For the autocorrelation function based on

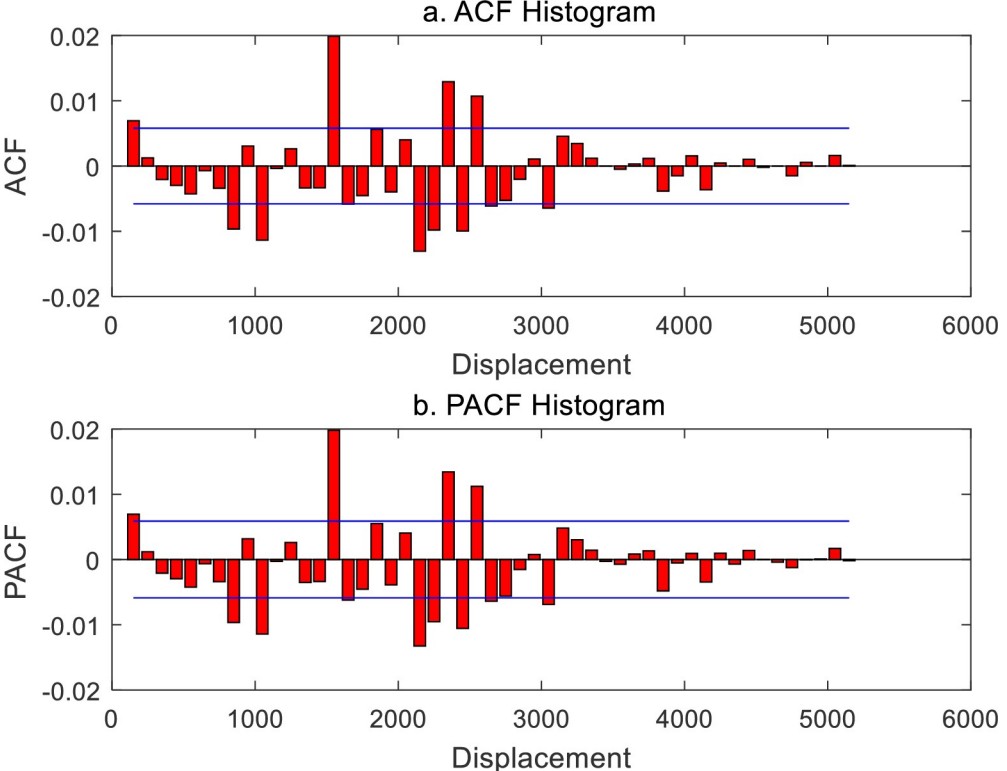

**Fig 5. Spatial autocorrelation function and partial autocorrelation function of Chinese cities based on conventional Moran's index and correlation density (2010).**

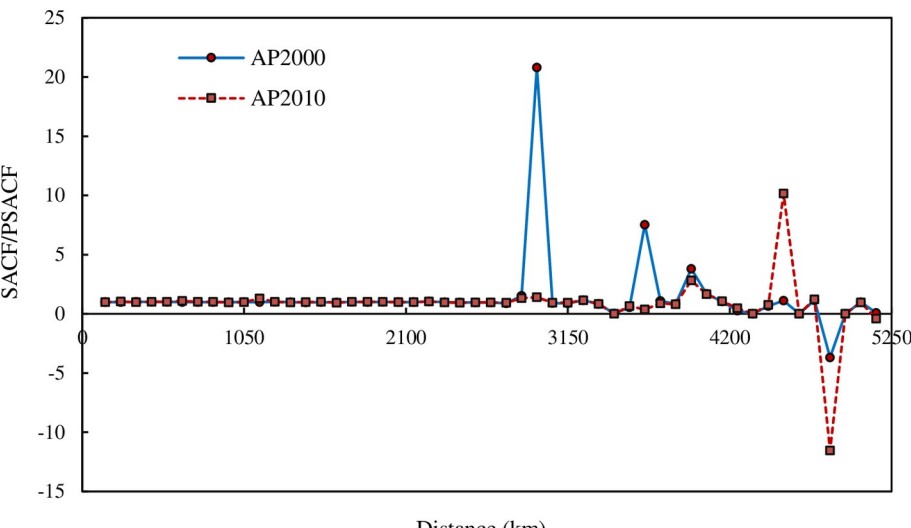

**Fig 6. The ratios of SACF to PSACF based on correlation density for the main cities of China.** (**Note**: Inside the scaling ranges of spatial correlation dimension, the ratios of spatial autocorrelation function to the partial spatial autocorrelation function are stable; In contrast, outside the scaling range, the ratio curves fluctuate significantly. From 2000 to 2010, the scaling range extended from about 2850 to 3350 km.).

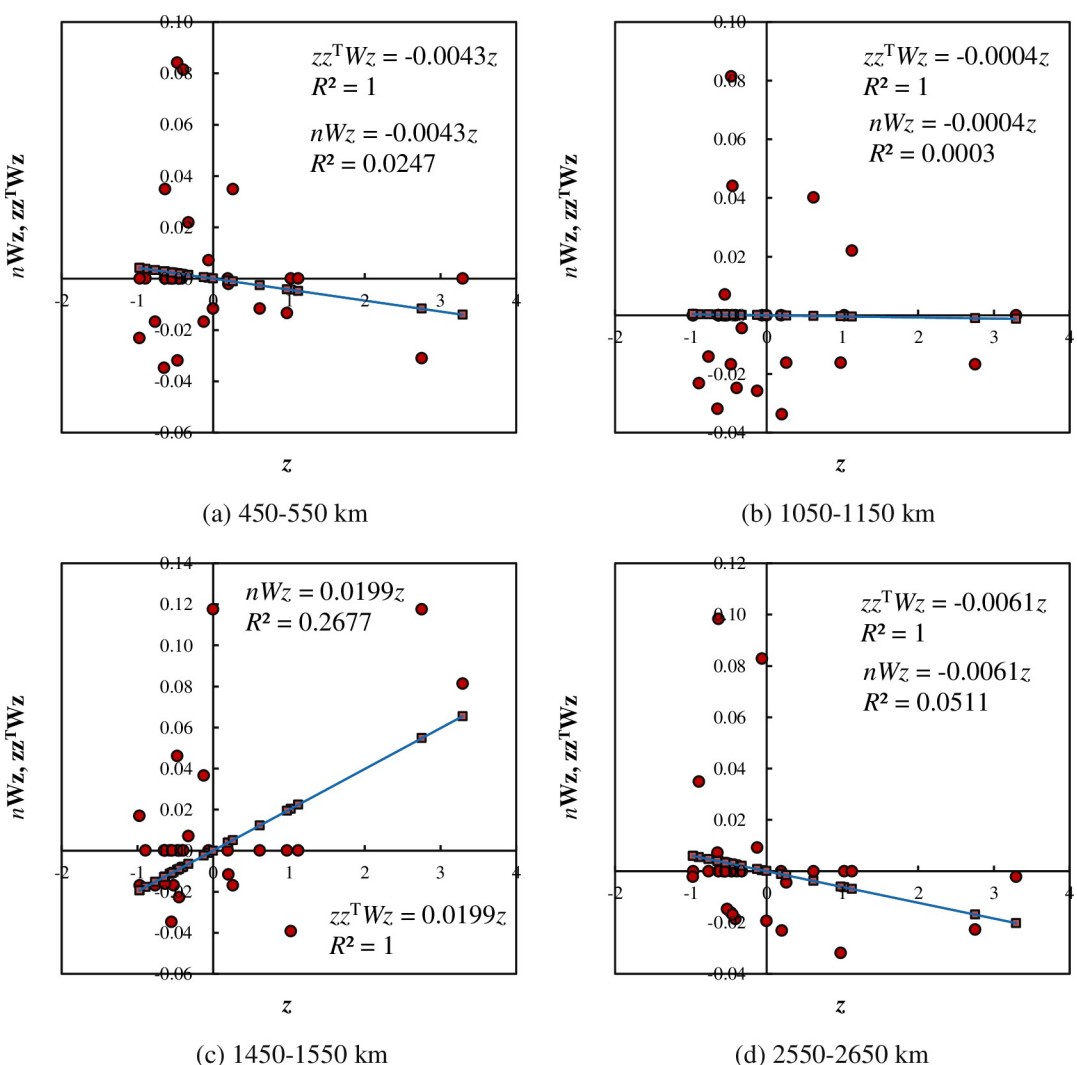

**Fig 7. The canonical Moran's scatterplots of spatial autocorrelation based on correlation density function for the main cities of China (examples for 2010). (Note**: The scatter points are based on the inner product correlation, $z^{\mathrm{T}}zWz = Iz$, and the relation is Eq (14). The trend line is based on the outer product correlation, $zz^{\mathrm{T}}Wz = Iz$, and the relation is Eq (16). The Moran's index difference values are as follows. (a) For $450 < r \leq 550$, $\Delta I = -0.0043$; (b) For $1050 < r \leq 1150$, $\Delta I = -0.0004$; (c) For $1450 < r \leq 1550$, $\Delta I = 0.0199$; (d) For $2550 < r \leq 2650$, $\Delta I = 0.0061$.).

Geary's coefficient, the power law relation is as below

$$C(r) = \frac{n-1}{n}[e^{\mathrm{T}}W(r)z^2 - I(r)] = C_0 r^a, \tag{43}$$

where $C_0$ refers to the proportionality coefficient, and $a$ to a scaling exponent. For the Getis-Ord's index, the power law relation is as follows

$$G(r) = y^{\mathrm{T}}W(r)y = G_0 r^b, \tag{44}$$

in which $G_0$ refers to the proportionality coefficient, and $b$ to a scaling exponent. The empirical analyses show that both the spatial cumulative autocorrelation function based on Geary's coefficient and that based on Getis-Ord's index follow power law if scaling range is taken into

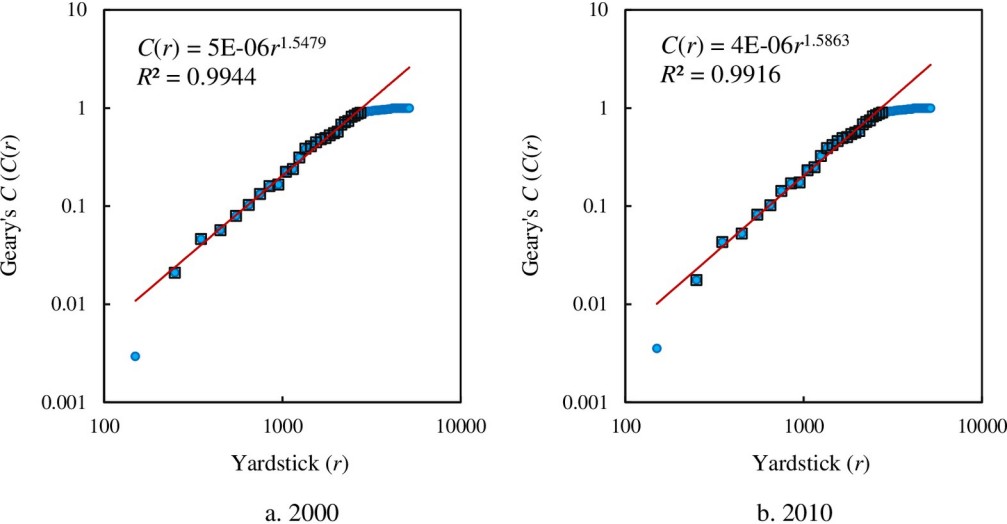

**Fig 8. The scaling relations for the spatial autocorrelation function based on cumulative correlation and Geary's coefficient. Note:** The solid dots represent all points of spatial autocorrelation functions, and the hollow blocks represent the points within the scaling range. The scaling range comes between 250 and 2750 km.

account (Figs 8 and 9). For Geary's coefficient, the scaling ranges from 250 to 2750 km. The scaling exponent is about 1.5479 in 2000 and about 1.5863 in 2010 (Fig 8). For Getis-Ord's index, the scaling range come between 150 and 2750 km. The scaling exponent is about 1.5811 in 2000 and about 1.4986 in 2010 (Fig 9).

Using a difference function, we can transform the cumulative autocorrelation functions based on Geary's coefficient and Getis-Ord's index into density autocorrelation functions. For Geary's coefficient, the formula for the autocorrelation function based on density correlation

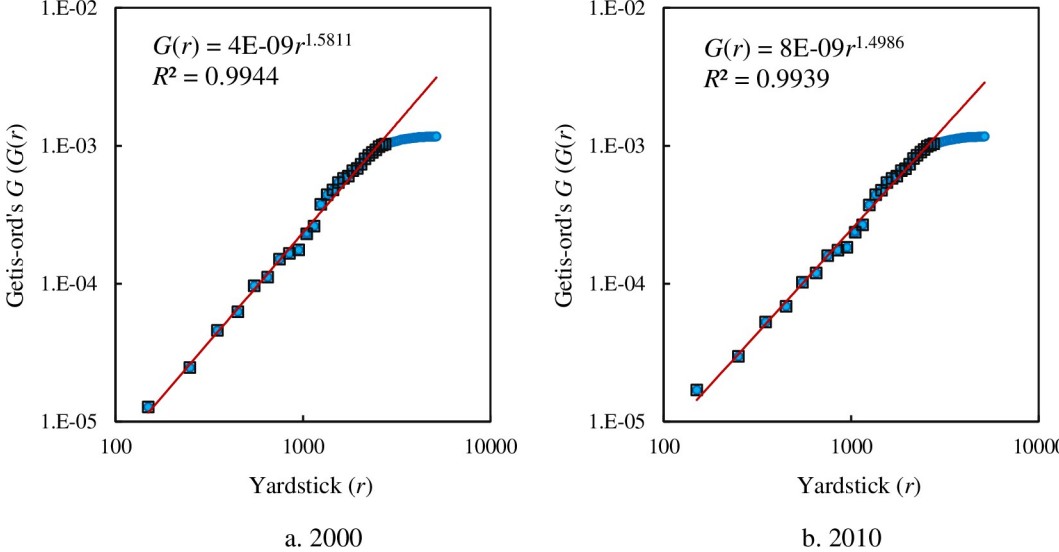

**Fig 9. The scaling relations for the spatial autocorrelation function based on cumulative correlation and Getis-Ord's index. Note:** The solid dots represent all points of spatial autocorrelation functions, and the hollow blocks represent the points within the scaling range. The scaling range comes between 150 and 2750 km.

is as follows

$$C_d(r) = \begin{cases} C(r_k), & k = 1 \\ \Delta C(r_k) = C(r_k) - C(r_{k-1}), & k > 1 \end{cases}, \qquad (45)$$

For Getis-Ord's coefficient, the formula for the autocorrelation function based on density correlation is as below

$$G_d(r) = \begin{cases} G(r_k) = y^{\mathrm{T}} W(r_k) y, & k = 1 \\ \Delta G(r_k) = G(r_k) - G(r_{k-1}), & k > 1 \end{cases}, \qquad (46)$$

In Eqs (45) and (46), the distance $r$ is discretized as $r_k = r_0 + ks$, in which $k = 1, 2, 3, \ldots, m$ represents natural numbers, $s$ refers to step length, and $r_0$ is a constant.

The autocorrelation function based on density correlation can reflect the scaling range of spatial dependence from another angle of view. The density correlation function curves are a random fluctuation curves, and the autocorrelation function curve based on Geary's coefficient looks like that based on Getis-Ord's index (Figs 10 and 11). Within the scaling range, spatial correlation changes greatly with distance, while outside the scaling range, spatial correlation changes slightly over distance. This feature is similar to the change trend of the density autocorrelation function based on Moran's index. The scaling range suggested by Geary's coefficient and Getis-Ord's index corresponds to that suggested by Moran's index (150 or 250 km -2750 or 3350 km).

## 4 Discussion

This work concerns two aspects of innovation in spatial modeling and analysis. First, in theory, the idea of spatial scaling is introduced into spatial autocorrelation modeling. Conventional spatial autocorrelation analysis is based on a fixed distance threshold and characteristic scales. Moran's $I$ is actually an eigenvalue of the generalized spatial correlation matrix. If and only if an eigenvalue bears no scale dependence, it can serve as a characteristic length in spatial analysis. Unfortunately, in many cases, Moran's $I$ depends on the spatial measurement scale. In this paper, spatial autocorrelation modeling is based on variable distance threshold and scaling. Moran's $I$ can be associated with the spatial correlation dimension. Second, in methodology, the spatial correlation coefficients were generalized to spatial autocorrelation functions and partial autocorrelation functions. Using these functions, we can perform analyses of the spatial dynamics of complex geographical systems. As we know, the modeling methods of ACF and PACF as well as the related spectrums have been developed for time series analysis. A time series is in fact a 1-dimensional variable based on ordered point sets. The methods for time series analysis can be applied to 1-dimensional spatial series based on isotropic ordered spatial point sets, but the methods cannot be directly generalized to the 2-dimensional spatial data based on anisotropic random spatial point sets (Table 3). A correlogram is a basic way of illustrating ACF and PACF in time series analysis. It is natural for this tool to be introduced into spatial autocorrelation analysis based on variable distance. However, the development of spatial autocorrelation function analysis is systems engineering. The methodology cannot be completely represented by correlograms and autocorrelation coefficients based on variable distance. Developing spatial autocorrelation functions relies heavily on three necessary conditions. First, introduction of a spatial displacement parameter into the spatial contiguity matrix. This step is easy to do, and, as mentioned above, many scholars have already done so [9,11–13]. The key is to select a proper distance decay function. Second, definition of the spatial contiguity matrix. This seems to be an easy problem to solve, but it is not. This involves the

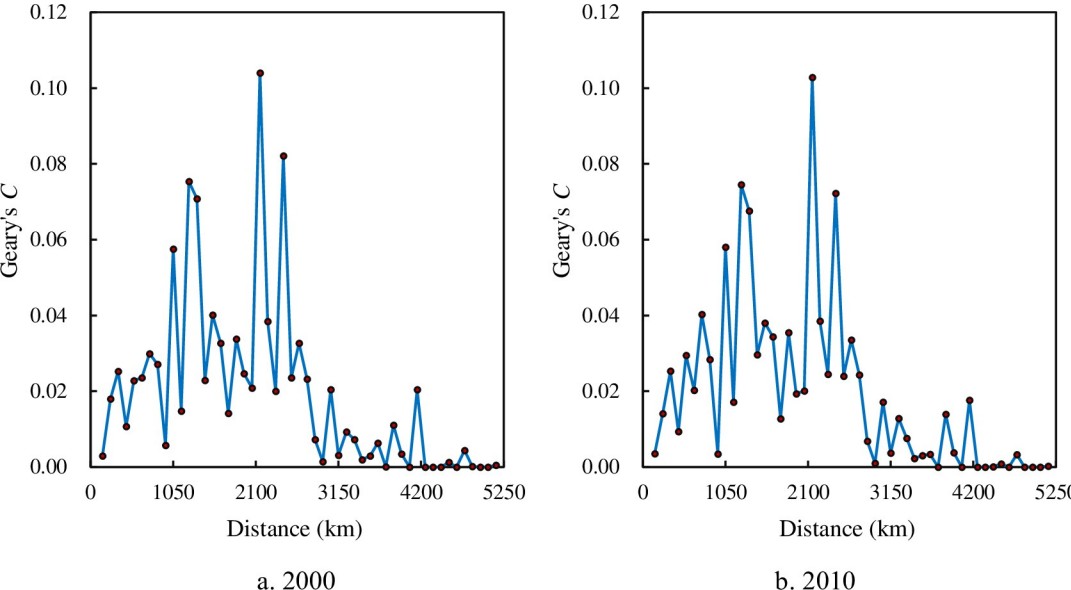

**Fig 10. The curves of Geary's *C* based on correlation density for the main cities of China.** (**Note**: Inside the scaling ranges of spatial correlation dimension, the curves of Geary's *C* fluctuate sharply. From 2000 to 2010, the curve shapes have no significant change.).

treatment of diagonal elements of the spatial weight matrix. Third, conversion of the spatial contiguity matrix to a weight matrix. If and only if the spatial contiguity matrix is normalized to yield a weight matrix, the key step can be revealed clearly, that is, the sum used to normalize the spatial contiguity matrix does not change over the spatial displacement parameter. Otherwise, the spatial autocorrelation functions cannot correspond to the temporal autocorrelation functions of time series analysis.

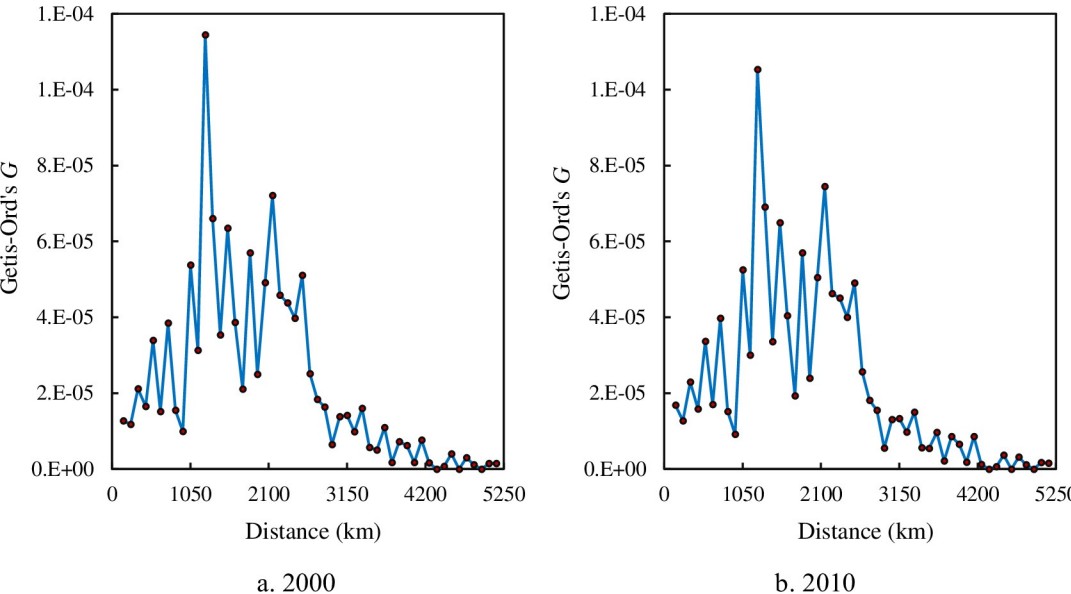

**Fig 11. The curves of Getis-Ord's *G* based on correlation density for the main cities of China.** (**Note**: Inside the scaling ranges of spatial correlation dimension, the curves of Getis-Ord's *G* fluctuate significantly. From 2000 to 2010, the curve shapes have slight change.).

**Table 3. A comparison of the analytical processes of autocorrelation functions and related methods.**

| Domain | Dimension | Object | ACF | PACF | Spectral analysis |
|---|---|---|---|---|---|
| **Time domain** | 1-dimension time (one independent direction) | Time series | Temporal ACF analysis and auto-regressive process | Temporal PACF analysis and auto-regressive process | Power spectrum |
| **Spatial domain** | 1-dimension space (one independent direction) | Isotropic ordered spatial series | Spatial ACF analysis and auto-regressive process | Spatial PACF analysis and auto-regressive process | Wave spectrum |
| | 2-dimension space (two independent directions) | Anisotropic random spatial series | To be developed | To be developed | To be developed |

Spatial autocorrelation analysis has gone through two stages. The first stage is reflected in biometrics. At this stage, spatial autocorrelation measurements are mainly used as an auxiliary means of traditional statistical analysis. The prerequisite or basic guarantee of statistical analysis is that the sample elements are independent of each other [8]. To measure the independence of spatial sampling results, Moran's index was presented by analogy with Pearson's product-moment correlation coefficient and the autocorrelation coefficient in time series analysis [36,37]. Moran's *I* is based on spatial populations (universes) rather than spatial samples [14]. As an addition, Geary's coefficient was proposed by analogy with the Durbin-Watson statistic [32], and this index is for spatial sample analysis [14]. The second stage is reflected in human geography. At this stage, spatial autocorrelation becomes one of the leading tools of geospatial modeling and statistical analysis. In the period of the geographical quantitative revolution (1953–1976), autocorrelation measurements were introduced into geography [7,38]. Geographers have found that few types of geospatial phenomena do not have spatial correlations, so traditional statistical analysis often fails in geographical research [8,38]. Geographers changed their thinking and decided to develop a set of analytical processes based on spatial autocorrelation [8,24,28,39–42]. A number of new measurements and methods emerged, including the Getis-Ord's index (Getis's *G*) [11,29], local Moran's indexes and Moran's scatterplot [33,43], spatial filtering [24], and spatial auto-regression modeling [44,45]. At the same time, spatial autocorrelation analysis continued to develop in biometrics and ecology [9,10,12,22,46–48]. At present, autocorrelation analysis seems to enter the third stage. Based on spatial autocorrelation measures and analytical processes, spatial statistics have been developed rapidly and applied to many areas [21,49–56]. However, if the spatial statistics are confined to autocorrelation coefficients and related measures, it will be difficult to further extend the applications and functions of spatial modeling and analysis.

Spatial autocorrelation methods open new ways of geographical statistical analysis under the conditions of existing inherent correlation among spatial sampling points. In particular, it lays the foundation for spatial autoregressive modeling. However, growing evidence shows that the measurement values of spatial autocorrelation indexes depend on size, shape, and spatial scales of geographical systems [8,9,11,12]. At least two approaches can be used to solve this problem. One is to make spatial scaling analysis based on spatial autocorrelation indexes, and the other is develop spatial autocorrelation function analysis. One basic method of developing spatial autocorrelation functions is to make use of variable distance. Based on the variable distance defined in spatial contiguity matrices, spatial correlation function, structure function, spatial correlogram, spline correlogram, and so on, have been introduced into spatial autocorrelation processes [8,9,12]. Spatial correlogram is just a result from analogy with the correlation function histogram in time series analysis. Among various methods of spatial analyses based on variable distance, the structure function advanced by Legendre and Legendre [12] looks like the spatial autocorrelation function developed in this work. However, there is an essential difference between structure function and autocorrelation function. A comparison

**Table 4. The differences and similarities between structure function and spatial autocorrelation function.**

| Item | Legendres' work | Work in this paper |
|------|-----------------|--------------------|
| Objective | Finding typical autocorrelation index | Find spatial scaling and relations to fractal dimension |
| Basic postulate | Characteristic scale | Scaling invariance |
| Statistic hypothesis | Gaussian distribution | Pareto distribution |
| Spatial contiguity definition | Kronecker's delta | Heaviside function (step function) |
| Distance conversion | Metric variable → Rank variable → Categorical variable | Metric variable → Categorical variable |
| Spatial weight matrix | (1) Based on variable mean; (2) Diagonals are zeros | (1) Based on fixed mean; (2) Diagonals are zeros or ones |
| Spatial correlation | Correlation density | Correlation cumulation |
| Measurement method | Variable distance | Variance distance |
| Measurement result | Autocorrelation coefficients | Autocorrelation and partial autocorrelation coefficients |
| Modeling result | Structure function | Autocorrelation function partial autocorrelation function |
| Representation way | Correlogram | Correlogram |
| Function | Spatial structure analysis based on characteristic scale | Spatial dynamics analysis based on scaling idea |

can be drawn by tabulating the similarities and differences between structure function and spatial autocorrelation function (Table 4). In short, the structure function is based on the idea of characteristic scales, while the spatial autocorrelation function is associated with scaling analysis for geographical systems. In fact, the variable distance can be employed to find the characteristic scale of spatial autocorrelation processes [8].

The empirical analysis results demonstrate that the 2-dimensional spatial autocorrelation coefficients and the related statistics can be generalized to 2-dimensional spatial autocorrelation functions and related functions. A preliminary framework of spatial analysis based on autocorrelation functions was put forward. The main contributions of this study can be outlined as three aspects. First, construction of 2-dimensional spatial autocorrelation functions. Based on Moran's index and the relative staircase function with a spatial displacement parameter, two sets of spatial autocorrelation functions are constructed. Second, definition of partial spatial autocorrelation functions. By means of the Yule-Walker recursive equation, the calculation approach of partial autocorrelation functions is proposed. Third, generalization of the spatial autocorrelation functions. The 2-dimensional spatial autocorrelation function are generalized to Geary's coefficient and Getis' index and the extended autocorrelation functions are established. Moreover, the spatial autocorrelation analysis based on characteristic scales is generalized to that based on scaling. The concept of scaling was associated with spatial autocorrelation [10]. However, the substantial research on scaling in spatial autocorrelation process has not been reported. The main mathematical expressions can be tabulated for comparison (Table 5). The significance of developing this mathematical framework for spatial autocorrelation lies in three respects. First, spatial information mining of geographical systems. The spatial autocorrelation functions can be used to reveal more geographical spatial information and express more complex dynamic processes than the spatial autocorrelation coefficients. Second,

**Table 5. Collections of two types of spatial autocorrelation functions and the extended results.**

| Type | Base | Standard SACF ($i \neq j$) | Generalized SACF ($i = j$) |
|------|------|-----------------------------|------------------------------|
| Basic functions | Moran's $I$: SACF | $I(r) = z^T W(r) z$ | $I^*(r) = z^T W^*(r) z$ |
|  | PSACF | $J(r) = f(I(r))$ | $J^*(r) = f(I^*(r))$ |
| Extended functions | Geary's $C$ | $C(r) = \frac{n-1}{n}[e^T W(r) z^2 - I(r)]$ | $C^*(r) = \frac{n-1}{n}[e^T W^*(r) z^2 - I^*(r)]$ |
|  | Getis's $G$ | $G(r) = y^T W(r) y$ | $G^*(r) = y^T W^*(r) y$ |
| Difference | SWM | $W(r) = V(r)/V_0(r)$ | $W^*(r) = V(r)/(n(n\text{-}1))$ |

foundation of scale and scaling analysis. If a geographical system bears characteristic scales, the spatial autocorrelation functions can be used to bring to light the characteristic length; if a geographical system has no characteristic scale, the spatial autocorrelation functions can be employed to perform scaling analysis. Third, future development of spectral analysis. Autocorrelation functions and power/wave spectral density represents two different sides of the same coin. Based on the spatial autocorrelation functions, the method of 2-dimensional spectral analysis can be developed for geographical research.

The new development of a theory or a method always gives rise to a series of new problems. New problems will lead to further exploration about the theory or the method. The main shortcomings of this work are as follows. First, the local spatial autocorrelation functions have not been taken into consideration. Moran's index, Geary's coefficient, and Getis' index can be used to measure local spatial autocorrelation. However, local spatial coefficients have not been generalized to local spatial autocorrelation functions. Second, the auto-regression models have not been built. Autocorrelation and auto-regression represent two different sides of the same coin. How can the auto-regression models, which can give the partial autocorrelation coefficients, be conducted? This is a pending question. Third, the case study is based on 29 provincial capital cities rather than a system of cities based on certain size threshold. The system of provincial capital cities are in the administrative sense instead of pure geographical sense. This type of spatial sample can be used to generate an example to illustrate the research method. If we perform a spatial analysis of Chinese cities, we should make a spatial sampling according to certain scale threshold.

## 5 Conclusions

A new analytical framework based on a series of spatial autocorrelation functions has been demonstrated with its mathematical derivation. A case study is presented to show how to make use of this analytical process. Next, we further improve the related spatial analytical methods based on spatial autocorrelation functions, including spatial cross-correlation functions, spatial auto-regression modeling, and spatial wave-spectral analysis. The main points can be summarized as follows. First, a new spatial analytical process can be developed by spatial autocorrelation functions based on the relation staircase function. By introducing a spatial displacement parameter into spatial weight functions, we can transform the spatial autocorrelation coefficients such as Moran's index into spatial autocorrelation functions on the analogy of the corresponding methods in time series analysis. An autocorrelation function is a parameter set comprising a series of autocorrelation coefficients. A spatial autocorrelation coefficient can be used to characterize the simple spatial correlation and structure, while a spatial autocorrelation function can be employed to describe the complex spatial correlation and dynamics. Second, partial spatial autocorrelation functions can be used to assist spatial autocorrelation function analysis. Using the Yule-Walker recursive equation, we can convert the spatial autocorrelation function based on Moran's index into partial spatial autocorrelation functions. Spatial autocorrelation functions reflect both direct and indirect spatial autocorrelation processes in a system, while partial spatial autocorrelation functions can be employed to display the pure direct autocorrelation process. Third, the spatial autocorrelation function can be extended by means of more spatial autocorrelation measurements. The spatial autocorrelation functions can be generalized to the autocorrelation functions based on Geary's coefficient and Getis-Ord' index. Different autocorrelation functions have different uses in spatial analysis. Using the spatial autocorrelation functions, we can mine more geographical spatial information, seek the characteristic scales for spatial modeling and quantitative analysis, or reveal the hidden scaling in complex geographical patterns and processes.

## Supporting information

**S1 File. Datasets of urban population and railway distances in 2000 for calculating spatial autocorrelation and partial autocorrelation functions.** This file contains the original or preliminarily processed data of 2000 used in this paper. It provides two complete processes of computing spatial autocorrelation function (ACF) and partial autocorrelation function (PACF).
(XLSX)

**S2 File. Datasets of urban population and railway distances in 2010 for calculating spatial autocorrelation and partial autocorrelation functions.** This file contains the original or preliminarily processed data of 2010 used in this paper.
(XLSX)

**S3 File. Four Matlab programs for computing the spatial autocorrelation and partial autocorrelation function analysis based on Moran's index.** It provides four Matlab programs for calculating spatial autocorrelation function (ACF) and partial autocorrelation function (PACF). Among the four programs, two are significant: one is based on diagonal elements and variable weights, and the other is based on zero diagonal elements and fixed weights. Readers can employ the programs to carry out spatial ACF and PACF analyses by substituting the author's data with their own data.
(M)

## Acknowledgments

I would like to thank the two anonymous reviewers and Dr. Stéphane Dray whose interesting and constructive comments were very helpful in improving the quality of this paper.

## Author Contributions

**Conceptualization:** Yanguang Chen.

**Data curation:** Yanguang Chen.

**Formal analysis:** Yanguang Chen.

**Funding acquisition:** Yanguang Chen.

**Investigation:** Yanguang Chen.

**Methodology:** Yanguang Chen.

**Project administration:** Yanguang Chen.

**Resources:** Yanguang Chen.

**Software:** Yanguang Chen.

**Supervision:** Yanguang Chen.

**Validation:** Yanguang Chen.

**Visualization:** Yanguang Chen.

**Writing – original draft:** Yanguang Chen.

**Writing – review & editing:** Yanguang Chen.

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
