## [Decision Letter · Decision Letter 0]

5 Jan 2021

PONE-D-20-34327

An Analytical Process of Spatial Autocorrelation Functions Based on Moran's Index

PLOS ONE

Dear Dr. Chen,

Thank you for submitting your manuscript to PLOS ONE. After careful consideration, we feel that it has merit but does not fully meet PLOS ONE’s publication criteria as it currently stands. Therefore, we invite you to submit a revised version of the manuscript that addresses the points raised during the review process.

We look forward to receiving your revised manuscript.

Kind regards,

Bailang Yu

Academic Editor

PLOS ONE

Journal Requirements:

2.Thank you for stating the following in the Acknowledgments Section of your manuscript:

"This research was sponsored by the National Natural Science Foundation of China (Grant No.

41671167. See: http://isisn.nsfc.gov.cn/egrantweb/). The support is gratefully acknowledged"

Additional Editor Comments (if provided):

Although there is a major objection from one of the reviewers, other two reviewers as well as myself think this is an interesting paper. I suggest the authors to revise the manuscript accordingly.

Reviewers' comments:

Reviewer's Responses to Questions

**Comments to the Author**

1. Is the manuscript technically sound, and do the data support the conclusions?

Reviewer #1: Yes

Reviewer #2: Partly

Reviewer #3: Yes

2. Has the statistical analysis been performed appropriately and rigorously? 

Reviewer #1: Yes

Reviewer #2: N/A

Reviewer #3: Yes

3. Have the authors made all data underlying the findings in their manuscript fully available?

Reviewer #1: Yes

Reviewer #2: Yes

Reviewer #3: Yes

4. Is the manuscript presented in an intelligible fashion and written in standard English?

Reviewer #1: No

Reviewer #2: No

Reviewer #3: No

5. Review Comments to the Author

Reviewer #1: This is an interesting study that attempts to unite the commonly used metrics of spatial autocorrelation such as Moran's I and Gettis G by developing matrix algebra that unites the formulae for the metrics and that supports multiscale analysis. The case study is population in major Chinese cities, and it illustrates the approach well.

I feel that the claim to link spectral power analysis with spatial analysis is not new, and that a broader literature search could reveal more prior work. Also, the paper needs a thorough proof edit of the English, as it lacks definite articles, does not balance verb tenses, mixes singulars and plurals etc. I will upload a marked up PDF that I hope the authors can use to improve the manuscript.

Reviewer #2: see attached pdf

Reviewer #3: Spatial autocorrelation is one of the most important methods in quantitative analyses of geography. This research proposed a new analytical framework based on a series of spatial autocorrelation functions, which, I believe, has made significant contributions both theoretically and practically. However, before publication, I think some points about the empirical analysis need to be clarified or discussed.

1. Only 29 capital cities were included in the analysis. Why not perform the analysis at a city level? I think it might be more appropriate for reflecting “scale”.

2. I don’t think Haikou should be excluded because it was linked to other capital cities like Guangzhou by train even there is no real railway between Guangdong Province and Haikou.

3. Only functions based on Moran’s Index were reported in subsection 3.2. How about other indexes? Will the extent to which the value of r influence the functions vary by index?

Some other comments:

1. Table 2”: it is better to introduce the unit of r, km in the table.

2. I think professional English editing is necessary to ensure the writing is of a high standard.

6. PLOS authors have the option to publish the peer review history of their article (what does this mean?). If published, this will include your full peer review and any attached files.

Reviewer #1: No

Reviewer #2: **Yes: **Stéphane Dray

Reviewer #3: No

---

## [Decision Letter · Decision Letter 1]

18 Feb 2021

PONE-D-20-34327R1

An Analytical Process of Spatial Autocorrelation Functions Based on Moran's Index

PLOS ONE

Dear Dr. Chen,

Thank you for submitting your manuscript to PLOS ONE. After careful consideration, we feel that it has merit but does not fully meet PLOS ONE’s publication criteria as it currently stands. Therefore, we invite you to submit a revised version of the manuscript that addresses the points raised during the review process.

There are still some minor comments from two reviewers. I invite you to revise the manuscript accordingly.

We look forward to receiving your revised manuscript.

Kind regards,

Bailang Yu

Academic Editor

PLOS ONE

Additional Editor Comments (if provided):

There are still some minor comments from two reviewers.

Reviewers' comments:

Reviewer's Responses to Questions

**Comments to the Author**

1. If the authors have adequately addressed your comments raised in a previous round of review and you feel that this manuscript is now acceptable for publication, you may indicate that here to bypass the “Comments to the Author” section, enter your conflict of interest statement in the “Confidential to Editor” section, and submit your "Accept" recommendation.

Reviewer #1: (No Response)

Reviewer #3: All comments have been addressed

2. Is the manuscript technically sound, and do the data support the conclusions?

Reviewer #1: Yes

Reviewer #3: Yes

3. Has the statistical analysis been performed appropriately and rigorously? 

Reviewer #1: Yes

Reviewer #3: Yes

4. Have the authors made all data underlying the findings in their manuscript fully available?

Reviewer #1: Yes

Reviewer #3: Yes

5. Is the manuscript presented in an intelligible fashion and written in standard English?

Reviewer #1: Yes

Reviewer #3: Yes

6. Review Comments to the Author

Reviewer #1: The paper has been improved, but suffers from a large number of minor language errors. I have left some substantial comments embedded in the PDF file, which I have uploaded here. There is one section where I would expect some citations for statements that I do not believe are correct.

Reviewer #3: Lines 556-558:"It is hard to clarify the whole questions about spatial scaling in the autocorrelation processes by means of several paragraph of words, and the related problems will be discussed in a companion paper."

Lines 693,694: "Due to limitation of space,...."

I don't think "space" can be a reason if the author does believe that the problem should be fully addressed in one paper. I would suggest tha author rephrase the sentence.

7. PLOS authors have the option to publish the peer review history of their article (what does this mean?). If published, this will include your full peer review and any attached files.

Reviewer #1: No

Reviewer #3: No

---

## [Editor Report · Decision Letter 2]

22 Mar 2021

An Analytical Process of Spatial Autocorrelation Functions Based on Moran's Index

PONE-D-20-34327R2

Dear Dr. Chen,

We’re pleased to inform you that your manuscript has been judged scientifically suitable for publication and will be formally accepted for publication once it meets all outstanding technical requirements.

Kind regards,

Bailang Yu

Academic Editor

PLOS ONE

---

## [Editor Report · Acceptance letter]

6 Apr 2021

PONE-D-20-34327R2 

An Analytical Process of Spatial Autocorrelation Functions Based on Moran’s Index 

Dear Dr. Chen:

I'm pleased to inform you that your manuscript has been deemed suitable for publication in PLOS ONE. Congratulations! Your manuscript is now with our production department. 

Kind regards, 

on behalf of

Dr. Bailang Yu 

Academic Editor

PLOS ONE